# WT-MVSNet: Window-based Transformers for Multi-view Stereo

**Jinli Liao** [1,2*]   **Yikang Ding** [1*]   **Yoli Shavit** [3]   **Dihe Huang** [1]   **Shihao Ren** [1,2]

**Jia Guo** [2]   **Wensen Feng** [2†]   **Kai Zhang** [1,4†]

[1] Tsinghua University   [2] Huawei Technologies   [3] Bar-Ilan University
[4] Research Institute of Tsinghua, Pearl River Delta

## Abstract

Recently, Transformers have been shown to enhance the performance of multi-view stereo by enabling long-range feature interaction. In this work, we propose Window-based Transformers (WT) for local feature matching and global feature aggregation in multi-view stereo. We introduce a Window-based Epipolar Transformer (WET) which reduces matching redundancy by using epipolar constraints. Since point-to-line matching is sensitive to erroneous camera pose and calibration, we match windows near the epipolar lines. A second Shifted WT is employed for aggregating global information within cost volume. We present a novel Cost Transformer (CT) to replace 3D convolutions for cost volume regularization. In order to better constrain the estimated depth maps from multiple views, we further design a novel geometric consistency loss (Geo Loss) which punishes unreliable areas where multi-view consistency is not satisfied. Our WT multi-view stereo method (WT-MVSNet) achieves state-of-the-art performance across multiple datasets and ranks $1^{st}$ on Tanks and Temples benchmark.

## 1   Introduction

Multi-view stereo (MVS) estimates depth maps of multi-view calibrated images in order to perform dense 3D reconstruction. MVS solutions aim to find correspondences between pixels in a reference image and epipolar lines in source images in order to estimate consistent depth values. Recently, Transformers were proposed for enabling long-range feature matching between reference and source images [3], achieving impressive reconstruction quality. However, matching each pixel in reference and source images without epipolar geometry constraints incurs matching redundancy. A recent effort to perform attention-based matching along the epipolar lines of source images [32], suffers instead from sensitivity to inaccurate camera pose and calibration, which can in turn results to erroneous matching. Another key step in contemporary learned MVS methods is the regularization of cost volume, generated by stacking cost maps associated with respective depth hypotheses. Typically, 3D CNNs are employed along the feature channels to estimate a probability function over different depth values [7, 33]. Recently, shifted window Transformers were shown to enable local feature aggregation while maintaining long-range cross interaction, surpassing CNNs across different vision tasks [4, 11, 12, 14, 15]. Interestingly, while learned MVS methods aim to estimate the likelihood of depth hypotheses from multi-view feature consistency, they calculate the absolute error between ground truth and predicted depth expectation without geometrical consistency supervision [2, 7, 33].

In this work, we propose Window-based Transformers (WT) to address both local feature matching and global feature aggregation. In order to introduce epipolar constraints into attention-based feature

---

*Equal contribution.

†Corresponding author.

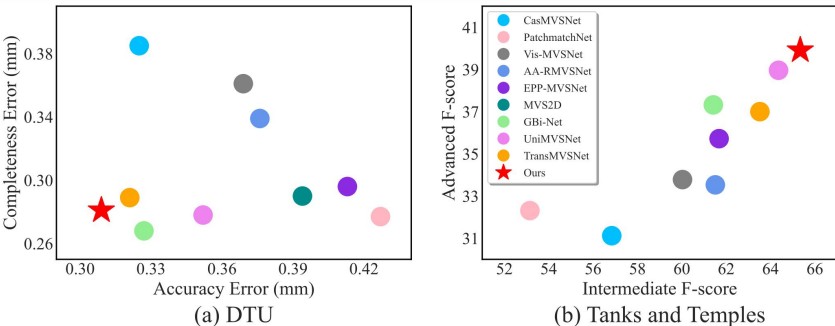

Figure 1: Comparison of performance with state-of-the-art learning-based MVS methods [3, 7, 16, 17, 19, 26, 27, 32, 37] on DTU dataset[5] (**lower is better**) and Tanks and Temples benchmark [9] (**higher is better**).

matching while maintaining robustness to camera pose and calibration inaccuracies, we develop a Window-based Epipolar Transformer (WET), which matches reference pixels and source windows near the epipolar lines. Motivated by the success of Transformers in visual feature aggregation, we further employ a window-based Cost Transformer (CT) for cost volume regularization. The CT is able to aggregate the global information within the whole cost volume and produces a much smoother and more complete probability volume. Finally, we extend the commonly used cross entropy loss (CE Loss) with our Geo Loss, which penalizes depth values that are not geometrically consistent across multiple views. Our WT-based method (WT-MVSNet) is evaluated on multiple MVS datasets. We show that it achieves significant and consistent improvement in terms of reconstruction accuracy and completeness and note that it ranks $1^{st}$ on the intermediate and advanced sets of Tanks and Temples benchmark [9].

In summary, our main contributions are as follows:

- We introduce a Window-based Epipolar Transformer (WET) for enhancing patch-to-patch matching between the reference feature and corresponding windows near epipolar lines in source features.
- We propose a window-based Cost Transformer (CT) to better aggregate global information within the cost volume and improve smoothness.
- We design a novel geometric consistency loss (Geo Loss) to supervise the estimated depth map with geometric multi-view consistency.
- Extensive experiments show that our method achieves state-of-the-art performance on multiple datasets. It ranks $1^{st}$ on the online Tanks and Temples benchmark.

## 2   Related works

### 2.1   Learning-based MVS

Over the past few years, learning-based MVS methods [7, 26, 34] have made significant progress in terms of accuracy, completeness, runtime and memory. A standard learning-based MVS pipeline typically consists of three modules: feature extraction, cost volume construction and cost volume regularization. MVSNet [33], which initially suggested this pipeline, computes a cost map for each depth hypothesis by warping the extracted features from several overlapping views using differentiable homography. It then builds a cost volume from stacked cost maps and regularizes it with a 3D UNet. This regularization process outputs a probability distribution over the different depth hypotheses. The depth map is estimated with the expectation or winner-take-all strategy of depth hypotheses, only with the supervision of ground truth depth map in reference view.

While being the first learning-based MVS method to achieve comparable results with traditional methods, the use of a 3D UNet for regularization carried high memory and runtime costs. Different methods were soon to follow, proposing alternative regularization approaches for mitigating this problem. Specifically, two main MVS categories have emerged: cascade-based 3D CNNs methods [2, 7, 31, 36, 37] and RNN-based methods [25, 27, 28, 30, 34]. Cascade-based methods leverage a multi-stage strategy to regularize the cost volume with a gradually narrower depth dimension. These

methods achieve significant improvement in memory, runtime and reconstruction quality compared to MVSNet, but struggle when trying to scale to high-resolution images. In order to address this limitation, RNN-based methods were proposed for performing recurrent regularization of cost maps. Such methods scale better in terms of memory but are much slower (due to the sequential nature of their regularization). However, neither of these kinds of methods utilize a global receptive field to perform regularization. Unlike these methods, our proposed CT first uses 3D window-based transformers for aggregating global information in cost regularization, making depth maps smoother and more complete.

## 2.2 Transformers and Attention for Feature Matching

Transformer [24], which was initially designed for natural language processing (NLP), has sparked the interest of the computer vision community due to its superior performance on vision challenges [1, 5, 14, 18, 20]. The ideology of Transformer has been used in the process of feature matching because of its inherent advantage in capturing global context information with an attention mechanism.

With alternating self-attention and cross-attention, SuperGlue [21] matches the sparse features via spatial relationships and the visual appearance of the keypoints. LoFTR [23] utilizes Transformer in a cascade structure to learn densely arranged and globally consented matching priors in ground-truth matches with interleaving self-attention and cross-attention for dense features matching. STTR [10] captures long-range global context information interaction between different features, leveraging alternating self-attention and cross-attention along epipolar lines.

TransMVSNet [3] first introduces Transformers into MVS task, which matches each reference pixel with the whole source images without using the epipolar constraints and achieves impressive results. MVS2D [32] uses the attention mechanism to perform point-to-line matching while ignoring the inaccurate camera calibrations. To solve such problems, we propose the WET that utilizes the window-based transformers for enhancing patch-to-patch matching, taking into account the epipolar constraint and the inaccurate camera calibration. Compared with TransMVSNet, our WET uses a much smaller matching space and costs less computation; compared with MVS2D, WET is more robust to the imperfect camera calibrations and poses.

## 3 Method

This section introduces the main contributions of our paper in detail. Firstly, we review the overall architecture of WT-MVSNet in Sec. 3.1, and then describe the three main contributions: Window-based Epipolar Transformer (WET) for global feature interaction in Sec. 3.2, Cost Transformer (CT) for cost regularization in Sec. 3.3 and geometric consistency loss (Geo Loss) in Sec. 3.4.

### 3.1 Network Overview

The overall architecture of WT-MVSNet is shown in Fig. 2. Given a reference image $\mathbf{I}_0 \in \mathbb{R}^{H \times W \times 3}$, several source images $\{\mathbf{I}_i\}_{i=1}^{N-1}$, their corresponding camera extrinsic matrixs $\{\mathbf{T}_i\}_{i=0}^{N-1}$ and intrinsic matrixs $\{\mathbf{K}_i\}_{i=0}^{N-1}$, as well as depth ranges $[d_{min}, d_{max}]$, WT-MVSNet aims to estimate the reference depth map $\mathbf{D}_0$ to reconstruct a dense 3D point cloud. Based on CasMVSNet [7], the first step is to extract the multi-scale features $\{\mathbf{F}_i\}_{i=0}^{N-1}$ of all input images via Feature Pyramid Network (FPN) [13] at 1/4, 1/2 and full image resolutions. To strengthen the global feature interaction within and across multi-view images, we propose a Window-based Epipolar Transformer (WET) to perform intra-attention and inter-attention alternately on the extracted features. Then we warp the transformed source features into reference view for building 3D cost volume $\mathbf{V}$ of $H \times W \times C \times D$, where $C$ and $D$ denote the number of feature channels and the number of depth candidates. After that, we use the proposed Cost Transformer (CT) to regularize $\mathbf{V}$ and produce the probability volume $\mathbf{P}$ of $H \times W \times D$, which aggregates the global cost information and can be used to generate the estimated depth. In the end, we utilize cross entropy loss (CE Loss) to supervise the probability volume and propose a geometric consistency loss (Geo Loss) to impose punishment in the area where the geometric consistency is not satisfied.

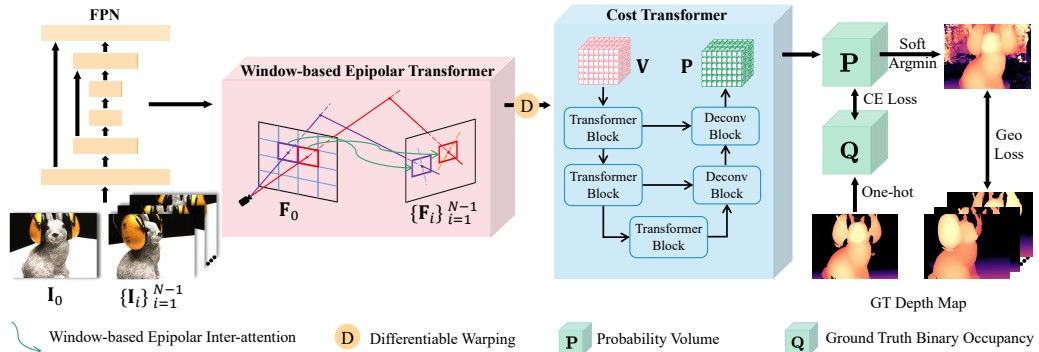

Figure 2: Overview of WT-MVSNet. Given multi-view images $\{\mathbf{I}_i\}_{i=0}^{N-1}$ and the corresponding camera extrinsic matrixs $\{\mathbf{T}_i\}_{i=0}^{N-1}$ and intrinsic matrixs $\{\mathbf{K}_i\}_{i=0}^{N-1}$, WT-MVSNet firstly extracts multi-scale features $\{\mathbf{F}_i\}_{i=0}^{N-1}$ of all input images. Then it uses Window-based Epipolar Transformer (WET) to strengthen the global feature interaction within and across $\{\mathbf{F}_i\}_{i=0}^{N-1}$. By performing global cost regularization using Cost Transformer (CT) on 3D cost volume $\mathbf{V}$, WT-MVSNet is able to produce a probability volume, which will be used to compute the estimated depth map by using winner-take-all strategy.

## 3.2 Window-based Epipolar Transformer

Most existing learning-based MVS methods that build cost volume directly via warping extracted features, result in lacking global context information. To address this problem, we introduce a Window-based Epipolar Transformer (WET) which reduces matching redundancy by using epipolar constraints. Since point-to-line matching is sensitive to erroneous camera calibration, we match windows near the epipolar lines. Sec. 3.2.1 introduces the preliminaries including Swin Transformer and intra-attention. Sec. 3.2.2 introduces the proposed Window-based Epipolar Inter-attention, which is the core of WET. Sec. 3.2.3 describes the overall architecture of WET.

### 3.2.1 Preliminaries

**Attention mechanism**    Swin Transformer [14] proposes a hierarchical feature representation with only linear computational complexity, achieving excellent performance in a variety of computer vision tasks. A Swin Transformer block contains window-based multi-head self attention (W-MSA) and shifted window-based multi-head self attention (SW-MSA), which can be formulated as:

$$\hat{\mathbf{z}}^l = W\text{-}MSA(LN(\mathbf{z}^{l-1})) + \mathbf{z}^{l-1}, \tag{1}$$

$$\mathbf{z}^l = MLP(LN(\hat{\mathbf{z}}^l)) + \hat{\mathbf{z}}^l, \tag{2}$$

$$\hat{\mathbf{z}}^{l+1} = SW\text{-}MSA(LN(\mathbf{z}^l)) + \mathbf{z}^l, \tag{3}$$

$$\mathbf{z}^{l+1} = MLP(LN(\hat{\mathbf{z}}^{l+1})) + \hat{\mathbf{z}}^{l+1}, \tag{4}$$

where $LN$ and $MLP$ denote the LayerNorm and Multilayer Perception. $\hat{\mathbf{z}}^l$ and $\mathbf{z}^l$ are the outputs of $(S)W\text{-}MSA$ and $MLP$ of the $l^{th}$ block. Swin Transformer divides the feature into non-overlapping windows and groups as query $\mathbf{Q}$, key $\mathbf{K}$ and value $\mathbf{V}$. The feature-wise similarity is extracted by the dot product of $\mathbf{Q}$ and $\mathbf{K}$ corresponding to each $\mathbf{V}$. The formula for attention can be defined as:

$$Attention(\mathbf{Q}, \mathbf{K}, \mathbf{V}) = SoftMax\left(\mathbf{Q}\mathbf{K}^T/\sqrt{\mathbf{d}} + \mathbf{B}\right)\mathbf{V}, \tag{5}$$

where $\mathbf{d}$ presents the dimension of the query and key features. And $\mathbf{B}$ is the relative position bias.

**Intra-attention and Inter-attention**    When $\mathbf{Q}$ and $\mathbf{K}$ are taken from the same feature map, the attention layers will capture relevant information within the given feature map. On the contrary, when $\mathbf{Q}$ and $\mathbf{K}$ are taken from different feature maps, the attention layers will enhance context interaction between different views.

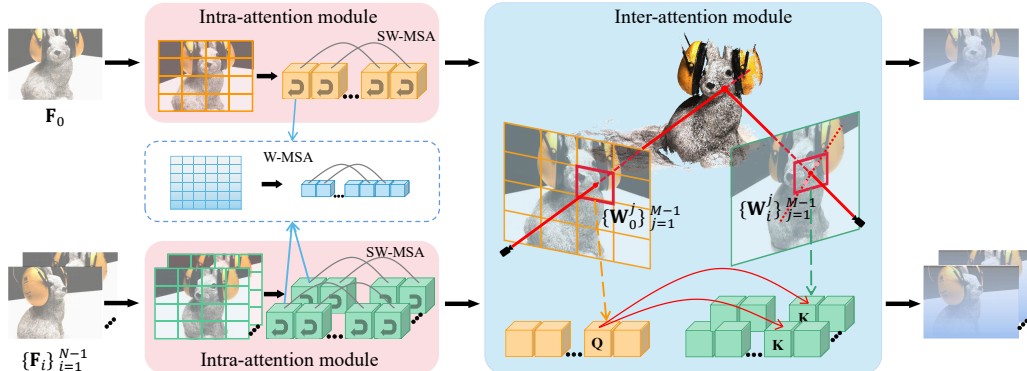

Figure 3: Illustration of WET. Given $\{\mathbf{F}_i\}_{i=0}^{N-1}$, WET divides $\mathbf{F}_i$ into non-overlapping windows and feed the flattened windows in Intra-attention module and compute W-MSA and SW-MSA sequentially. Then WET divides the reference feature into non-overlapping windows $\{\mathbf{W}_0^j\}_{j=0}^{M-1}$ and warps the corresponding center points to partition the corresponding windows on source features. By performing Window-based epipolar inter-attention, WET is able to enhance the feature interaction across views and improve the feature matching quality.

### 3.2.2 Window-based Epipolar Inter-attention

We perform inter-attention between $\mathbf{F}_0$ and each $\mathbf{F}_i$, while only updates $\mathbf{F}_i$ by following TransMVS-Net [3]. Specifically, we perform inter-attention between the pixels in $\mathbf{F}_0$ and corresponding patches along the epipolar lines in each $\mathbf{F}_i$ separately. The first step is to divide $\mathbf{F}_0$ into $M$ non-overlapping windows $\{\mathbf{W}_0^j\}_{j=0}^{M-1}$ with the same size $h_{win} \times w_{win}$, and warp the center points $\mathbf{p}_j$ of $\mathbf{W}_0^j$ into $\mathbf{F}_i$ via differentiable homography warping. The warped center points of $i^{th}$ source view $\mathbf{p}_i^j$ are:

$$\mathbf{p}_i^j = \mathbf{K}_i \left[ \mathbf{R} \left( \mathbf{K}_0^{-1} \mathbf{p}_0^j d \right) + \mathbf{t} \right], \tag{6}$$

where $\mathbf{R}$ and $\mathbf{t}$ denote the rotation and translation between reference view and source view. $d$ is the estimated depth value in the very first iteration of the coarsest stage. To perform inter-attention, we partition a window $\mathbf{W}_i^j$ around each $\mathbf{p}_i^j$ with the same size $h_{win} \times w_{win}$, in which the epipolar line of $\mathbf{p}_i^j$ in source feature goes through. Therefore, the inter-attention is able to strengthen long-range global context information interaction between the window of reference feature and the windows near the epipolar lines on source features.

### 3.2.3 WET Architecture

The architecture of the proposed WET is represented in Fig. 3. WET is mainly composed of intra-attention module and inter-attention module. In intra-attention module, the extracted features $\{\mathbf{F}_i\}_{i=0}^{N-1}$ are divided into non-overlapping windows. We flatten each window and feed it to W-MSA and SW-MSA sequentially. With only performing intra-attention within each partitioned window, the W-MSA cannot capture global context of the whole input feature. To solve this problem, we utilize SW-MSA and the shifted window partitioning strategy to enhance information interaction between different windows and obtain global context. In order to reduce matching redundancy and avoid erroneous camera pose and calibration, we perform Window-based Epipolar Inter-attention between reference and source views. In inter-attention module, we divide $\mathbf{F}_0$ into non-overlapping windows and warp each center point to partition the corresponding windows in source features. After flattening the partitioned windows, we compute the inter-attention between each window in $\mathbf{F}_0$ and the corresponding window in each $\mathbf{F}_i$ to transform and update $\mathbf{F}_i$ only.

### 3.3 Cost Transformer

In this section, we further explore the effects of different regularizations and find that the global receptive field has a significant impact on final performance. In practice, we propose a novel window-based Cost Transformer (CT) to aggregate the global information within cost volume. As shown in Fig. 4-(b), with the expansion of receptive field, the voxels of probability volume with the highest

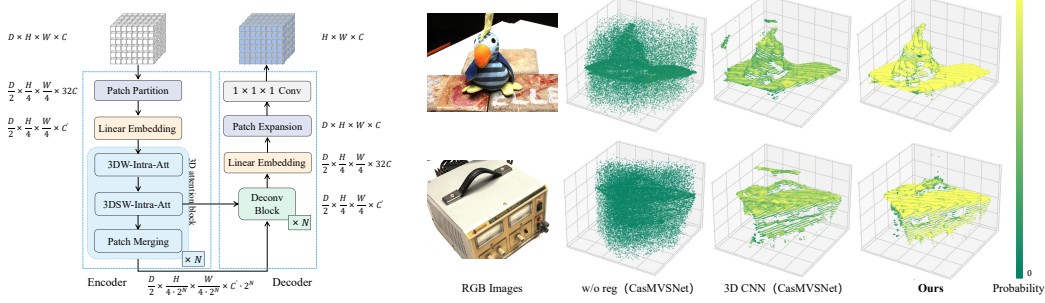

(a) The architecture of CT                    (b) Visualization comparison of probability volume

Figure 4: (a) The architecture of CT composes of encoder, decoder and skip connections. CT takes as input an initial cost volume and performs regularization with aggregating the global cost information. (b) Visualization of the voxel distribution of probability volume with the highest probability in depth dimension. Compared with 3D CNNs and no-regularization, CT is able to produce smoother and more complete probability volume, as well as higher confidence (yellow regions indicate higher probability, which is equivalent to higher confidence).

probability in depth dimension become smoother and more complete, as well as higher confidence. In comparison to 3D CNNs and no-regularization, our proposed CT produces probability volume with higher quality.

**3D attention**    As mentioned in Sec. 3.2.1, given a 2D feature $\mathbf{F}$, W-MSA and SW-MSA are able to capture the global context within $\mathbf{F}$. In the task of multi-view stereo, we are naturally faced with handling 3D features (cost volume $\mathbf{V}$). To leverage global receptive field in cost regularization, we extend the W-MSA and SW-MSA to 3D version. To do so, we generally flatten the 3D volume in spatial and depth dimensions. The following operations are similar to 2D attentions.

**CT Architecture**    The overall architecture of CT is shown in Fig. 4-(a), which consists of encoder, decoder and skip connections. Given an input cost volume $\mathbf{V}$, the encoder firstly divides $\mathbf{V}$ into non-overlapping 3D blocks, each 3D block will then be flattened from $2 \times 4 \times 4 \times C$ to $32C$. Besides, a linear embedding layer is applied to project the channel dimension $32C$ into $C'$ and produces the embedded cost volume $\mathbf{V}'$. After that, we further divide $\mathbf{V}'$ into non-overlapping 3D windows, and flatten each 3D window from $d_{win} \times h_{win} \times w_{win} \times C'$ to $d_{win}h_{win}w_{win} \times C'$. The flattened windows are then fed into $N$ 3D attention blocks, and each block consists of 3DW-Intra-Att, 3DSW-Intra-Att and patch merging layer. The patch merging layer is responsible for spatially down-sampling and increasing channel dimension. In the decoder, we leverage deconvolutions to restore the resolution. To decrease the loss of spatial information generated by the patch merging layer, we concatenate the shallow features and the deep features together with skip connections, which fuses the multi-scale features from the encoder with the decoder. Followed by a linear embedding layer and a patch expansion layer, the transformed $\mathbf{V}'$ remains the same as the $\mathbf{V}$ dimensions. In the end, a 3D convolution with $1 \times 1 \times 1$ kernel is applied to produce the final probability volume $\mathbf{P}$.

## 3.4  Loss Function

**Geometric consistency loss**    In general, the estimated depth maps are only supervised in reference view without using the multi-view consistency, which is usually used to filter out the outliers during the inference phase. We try to utilize such multi-view consistency in training phase and propose a novel geometric consistency loss (Geo Loss) to impose punishment in the area where geometric consistency is not satisfied. Firstly, we warp each pixel $\mathbf{p}$ in the estimated depth map $\mathbf{D}_0$ of reference view to obtain the corresponding pixel $\mathbf{p}'_i$ in $i^{th}$ neighboring source view by $\mathbf{p}'_i = \mathbf{T}_0^{-1}\mathbf{K}_0^{-1}\mathbf{D}_0(\mathbf{p})\mathbf{p}$, where $\mathbf{D}_0(\mathbf{p})$ denotes the depth value of pixel $\mathbf{p}$. In turn, we back-project $\mathbf{p}'_i$ into 3D space and then reproject it to the reference view as $\mathbf{p}''$:

$$\mathbf{p}'' = (\mathbf{K}_i\mathbf{T}_i\mathbf{p}'_i)/\mathbf{D}_i^{gt}(\mathbf{p}'_i), \tag{7}$$

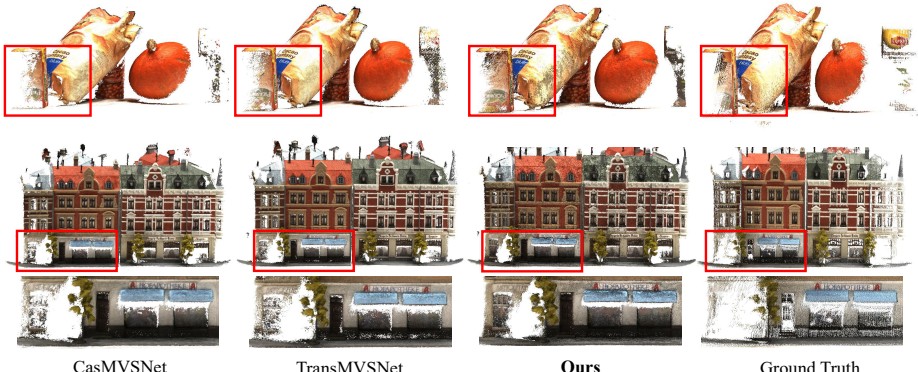

|        CasMVSNet        |        TransMVSNet        |        **Ours**        |        Ground Truth        |

Figure 5: Visualization comparison with state-of-the-art methods on DTU evaluation dataset [8] (scan15 and scan32). Compared with our baseline [7] and TransMVSNet [3], our WT-MVSNet obtains a more complete point cloud shown in the red bounding boxes.

Table 1: Quantitative results on DTU evaluation set [8]. The best results are in **Bold** and the second best figures are in underlined. Compared with existing methods, our method achieves superior performance in terms of overall error (**lower is better**).

| Method | Acc.($mm$) | Comp.($mm$) | Overall($mm$) |
|---|---|---|---|
| Gipuma [6] | **0.283** | 0.873 | 0.578 |
| COLMAP [22] | 0.400 | 0.664 | 0.532 |
| R-MVSNet [34] | 0.385 | 0.459 | 0.422 |
| AA-RMVSNet [27] | 0.376 | 0.339 | 0.357 |
| CasMVSNet [7] | 0.325 | 0.385 | 0.355 |
| EPP-MVSNet [16] | 0.413 | 0.296 | 0.355 |
| PatchmatchNet [26] | 0.427 | 0.277 | 0.352 |
| MVS2D [32] | 0.394 | 0.290 | 0.342 |
| UniMVSNet [19] | 0.352 | 0.278 | 0.315 |
| TransMVSNet [3] | 0.321 | 0.289 | 0.305 |
| GBi-Net [17] | 0.327 | **0.268** | 0.298 |
| **WT-MVSNet** | 0.309 | 0.281 | **0.295** |

where $\mathbf{D}_i^{gt}(\mathbf{p}_i')$ denotes the ground truth depth value of $\mathbf{p}_i'$. We define two reprojection errors as:

$$\xi_p = \left\| \mathbf{p} - \mathbf{p}'' \right\|_2, \tag{8}$$

$$\xi_d = \frac{1}{\mathbf{D}_0(\mathbf{p})} \left\| \mathbf{D}_0(\mathbf{p}'') - \mathbf{D}_0(\mathbf{p}) \right\|_1. \tag{9}$$

Thus the final Geo Loss $\mathcal{L}_{Geo}$ can be written by:

$$\mathcal{L}_{Geo} = \sum_{\mathbf{p} \in \{\mathbf{p}_v\} \bigcup \{\mathbf{p}_g\}} \Phi(\xi_p(\mathbf{p}) + \gamma \xi_d(\mathbf{p})), \tag{10}$$

where $\Phi$ denotes the Sigmoid function to normalize the combined reprojection errors with hyperparameters $\gamma$. $\{\mathbf{p}_v\}$ represents a set of valid spatial coordinates obtained by the valid mask map and $\{\mathbf{p}_g\}$ is a set of all pixels whose reprojection errors within the given thresholds, e.g. $\xi_p < \tau_1$ and $\xi_d < \tau_2$, where $\tau_1$ and $\tau_2$ is hyperparameters which decrease with the increase of the stage.

**Total loss** In summary, the loss function consists of cross entropy loss (CE Loss) and Geo Loss:

$$\mathcal{L} = \sum_{k=0}^{M-1} \lambda_1 \mathcal{L}_{CE} + \lambda_2 \mathcal{L}_{Geo}, \tag{11}$$

where $\lambda_1$ and $\lambda_2$ are the weights and $M$ is the number of the stages.

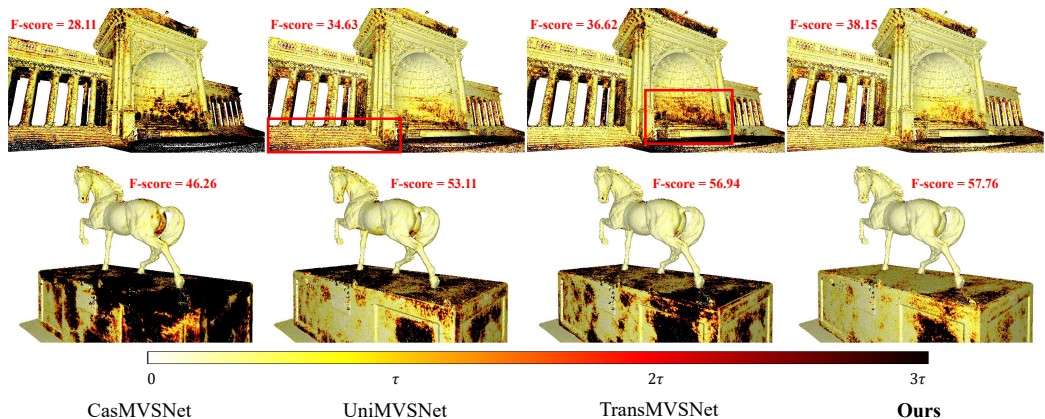

Figure 6: Comparison of reconstructed results with our baseline [7] and top-2 published models [19, 3] on Tanks and Temples benchmark [9]. The first row shows Recall on the advanced scene of Temple ($\tau = 15mm$); the second row shows Recall on the intermediate scene of Horse ($\tau = 3mm$).

## 4 Experiments

### 4.1 Implementation Details

We implement WT-MVSNet based on Pytorch, which is trained on DTU training set. Similar to CasMVSNet [7] with 3 stages of 1/4, 1/2 and full image resolutions, the corresponding depth interval decays by 0.25 and 0.5 from stage 1 to 3; the number of depth hypotheses is 48, 32 and 8 for each stage. When trained on DTU, the number of images is set to $N = 5$ and the image resolution is set to $512 \times 640$. We train our model using Adam for 16 epochs at a learning rate of 0.001, which decays by a factor of 0.5 after 6, 8, 12 epochs, respectively. We set combination coefficient $\gamma = 100.0$, the loss weights $\lambda_1 = 2.0$ and $\lambda_2 = 1.0$, the reprojection errors thresholds $\tau_1$ to 3.0, 2.0, 1.0 and $\tau_2$ to 0.1, 0.05, 0.01 at 3 resolutions. We train our model with the batch size being set to 1 on 8 Tesla V100 GPUs. WT-MVSNet typically takes 15 hours and occupies 13$GB$ of each GPU's RAM. Besides, the runtime and memory for inference are 0.786$s$ and 5221$MB$ respectively.

### 4.2 Datasets

There are several commonly used training and evaluation datasets for MVS as follows: (a) DTU is an indoor dataset that contains 128 scans with 49 views of 7 different lighting conditions under well-controlled laboratory conditions with a fixed camera trajectory. Following the setting of MVSNet [33], DTU dataset is split into 79 training scans, 18 validation scans, and 22 evaluation scans. (b) Tanks and Temples is a large-scale benchmark captured in realistic indoor and outdoor scenarios which contains an intermediate subset of 8 scenes and an advanced subset of 6 scenes. (c) BlendedMVS is a large-scale synthetic dataset for MVS training that contains cities, sculptures and small objects, which is split into 106 training scans and 7 validation scans.

### 4.3 Experimental Performance

**Evaluation on DTU** We train our model on DTU training set and test our model on DTU evaluation dataset [8] towards point clouds with the number of images as $N = 5$ and the image resolution as $864 \times 1152$. The quantitative results of DTU evaluation set are summarized in Tab. 1, where Accuracy and Completeness are a pair of official evaluation metrics. Accuracy is the percentage of generated point clouds matched in the ground truth point clouds, while Completeness measures the opposite. Overall is the mean of Accuracy and Completeness. Compared with the other methods, our proposed method shows its capability for generating denser and more complete point clouds on textureless regions, which is visualized in Fig. 5. All point clouds on DTU evaluation set are available from the Supplementary Material.

**Evaluation on Tanks and Temples** Before evaluating on Tanks and Temples benchmark [9] to validate the generalization of our model, we finetune WT-MVSNet on BlendedMVS [35] training

Table 2: Benchmarking results of F-score (**higher is better**) on the intermediate leaderboards of Tanks and Temples [9]. The best results are in **Bold** and the second best figures are in underlined. WT-MVSNet ranks $1^{st}$ among all the submitted methods on the intermediate leaderboards of Tanks and Temples benchmark (May. 19, 2022).

| Method | Int.Mean | Family | Francis | Horse | L.H. | M60 | Panther | P.G. | Train |
|---|---|---|---|---|---|---|---|---|---|
| COLMAP [35] | 42.14 | 50.41 | 22.25 | 26.63 | 56.43 | 44.83 | 46.97 | 48.53 | 42.04 |
| R-MVSNet [34] | 50.55 | 73.01 | 54.46 | 43.42 | 43.88 | 46.80 | 46.69 | 50.87 | 45.25 |
| PatchmatchNet [26] | 53.15 | 66.99 | 52.64 | 43.24 | 54.87 | 52.87 | 49.54 | 54.21 | 50.81 |
| CasMVSNet [7] | 56.84 | 76.37 | 58.45 | 46.26 | 55.81 | 56.11 | 54.06 | 58.18 | 49.51 |
| ACMM [29] | 57.27 | 69.24 | 51.45 | 46.97 | 63.20 | 55.07 | 57.64 | 60.08 | 54.48 |
| GBi-Net [17] | 61.42 | 79.77 | **67.69** | 51.81 | 61.25 | 60.37 | 55.87 | 60.67 | 53.89 |
| AA-RMVSNet [27] | 61.51 | 77.77 | 59.53 | 51.53 | 64.02 | 64.05 | 59.47 | 60.85 | 54.90 |
| EPP-MVSNet [16] | 61.68 | 77.86 | 60.54 | 52.96 | 62.33 | 61.69 | 60.34 | **62.44** | 55.30 |
| TransMVSNet [3] | 63.52 | 80.92 | 65.83 | 56.94 | 62.54 | 63.06 | 60.00 | 60.20 | **58.67** |
| UniMVSNet [19] | 64.36 | 81.20 | 66.43 | 53.11 | 63.46 | **66.09** | 64.84 | 62.23 | 57.53 |
| **WT-MVSNet** | **65.34** | **81.87** | 67.33 | **57.76** | **64.77** | 65.68 | 64.61 | 62.35 | 58.38 |

Table 3: Benchmarking results of F-score (**higher is better**) on the advanced leaderboards of Tanks and Temples [9]. Compared with all the published methods, WT-MVSNet achieves state-of-the-art performance on the advanced leaderboards of Tanks and Temples benchmark (May. 19, 2022).

| Method | Adv.Mean | Auditorium | Ballroom | Courtroom | Museum | Palace | Temple |
|---|---|---|---|---|---|---|---|
| COLMAP [35] | 27.24 | 16.02 | 25.23 | 34.70 | 41.51 | 18.05 | 27.94 |
| R-MVSNet [34] | 29.55 | 19.49 | 31.45 | 29.99 | 42.31 | 22.94 | 31.10 |
| CasMVSNet [7] | 31.12 | 19.81 | 38.46 | 29.10 | 43.87 | 27.36 | 28.11 |
| PatchmatchNet [26] | 32.31 | 23.69 | 37.73 | 30.04 | 41.80 | 28.31 | 32.29 |
| AA-RMVSNet [27] | 33.53 | 20.96 | 40.15 | 32.05 | 46.01 | 29.28 | 32.71 |
| ACMM [29] | 34.02 | 23.41 | 32.91 | **41.17** | 48.13 | 23.87 | 34.60 |
| EPP-MVSNet [16] | 35.72 | 21.28 | 39.74 | 35.34 | 49.21 | 30.00 | **38.75** |
| TransMVSNet [3] | 37.00 | 24.84 | **44.59** | 34.77 | 46.49 | **34.69** | 36.62 |
| GBi-Net [17] | 37.32 | **29.77** | 42.12 | 36.30 | 47.69 | 31.11 | 36.93 |
| UniMVSNet [19] | 38.96 | 28.33 | 44.36 | 39.74 | 52.89 | 33.80 | 34.63 |
| **WT-MVSNet** | **39.91** | 29.20 | 44.48 | 39.55 | **53.49** | 34.57 | 38.15 |

dataset to improve performance in real-world scenes using the original image resolution $576 \times 768$ and the number of images $N = 7$. Compared with the other methods, the performance of our model is introduced in Tab. 2 and Tab. 3, where F-score is defined as a harmonic mean of accuracy and completeness. Besides, Fig. 6 shows qualitative results on the scene Courtroom of advanced set and Horse of intermediate set. WT-MVSNet surpasses all existing learning-based MVS methods on intermediate set and outputs all submitted methods on advanced set, confirming its efficiency and generalizability. More point cloud results may be found in the Supplementary Material.

## 4.4 Ablation Study

As aforementioned, we adopt a new loss function and Transformer applied in feature matching and regularization. To demonstrate the effectiveness of our proposed modules, we conduct a quantitative ablation study. Our method is basically based on CasMVSNet [7], which applies feature correlation similar to PatchmatchNet [26] and L-1 regression loss to train. All experiments are conducted with the same parameters. With the quantitative performance shown in Tab. 4, we add CE Loss, Geo Loss, WET and CT based on baseline gradually. It can be seen from the table that our proposed modules improve in both accuracy and completeness. More extensive ablation experiments of our design choices are analyzed in the Supplementary Material.

Table 4: Ablation results (**lower is better**) with different components on DTU evaluation dataset [8].

| | module Settings | | | | Mean Distance | | |
|---|---|---|---|---|---|---|---|
| | CE Loss | Geo Loss | WET | CT | Acc.(mm) | Comp.(mm) | Overall(mm) |
| (a) | | | | | 0.351 | 0.311 | 0.331 |
| (b) | ✓ | | | | 0.342 | 0.307 | 0.325 |
| (c) | ✓ | ✓ | | | 0.336 | 0.302 | 0.319 |
| (d) | ✓ | ✓ | ✓ | | 0.328 | 0.288 | 0.308 |
| (e) | ✓ | ✓ | ✓ | ✓ | **0.309** | **0.281** | **0.295** |

# 5 Discussion

## 5.1 Limitations

In inter-attention module, the warped points are fixedly selected from reference feature, without considering the importance of the central point. Besides, the introduction of Transformer inevitably incurs high memory costs during the training phase and slows down the speed of inference.

## 5.2 Conclusion

In this paper, we propose WT-MVSNet, an end-to-end learning-based MVS network, for both local feature matching and global feature aggregation. We introduce a Window-based Epipolar Transformer (WET) to match windows near the epipolar lines which reduces matching redundancy and avoids erroneous camera pose and calibration. Besides, we present a novel Cost Transformer (CT) to replace 3D convolutions for aggregating global information within the cost volume. To better constrain the estimated depth maps from multiple views, we design a geometric consistency loss (Geo Loss) to impose punishment in the unreliable areas where multi-view consistency is not satisfied. Our WT-MVSNet achieves state-of-the-art performance across multiple datasets and ranks $1^{st}$ on challenging Tanks and Temples benchmark.

# 6 Acknowledgments

This work is supported by the Key-Area Research and Development Program of Guangdong Province (No.2020B0909050003), the Science and Technology Innovation Project of Shenzhen (JSGG20210802154807022) and the National Natural Science Foundation of China under Grants 61902415.

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
