# Supplementary Material for WT-MVSNet: Window-based Transformers for Multi-view Stereo

Jinli Liao [1,2*]   Yikang Ding [1*]   Yoli Shavit [3]   Dihe Huang [1]   Shihao Ren [1,2]

Jia Guo [2]   Wensen Feng [2†]   Kai Zhang [1,4†]

[1] Tsinghua University   [2] Huawei Technologies   [3] Bar-Ilan University
[4] Research Institute of Tsinghua, Pearl River Delta

## 1   Pathway of Window-based Epipolar Transformer

Due to Transformer limited by input image resolution, our proposed Window-based Epipolar Transformer (WET) only performs on features of $1/4$ raw resolution. Each pixel of reference feature corresponds to an epipolar line throughout the source feature. In inter-attention module, to better divide the corresponding windows of source feature, we iterate at $1/4$ resolution twice. The first iteration is to estimate depth map without WET. In the following iteration, we utilize the depth values to warp the center pixels of reference feature windows in order to partition corresponding windows in source features. To train WET with supervision at all stages, we design a transformed feature pathway that interpolates the feature map processed by WET to $1/2$ and full resolution and then adds to the corresponding feature map at the next stage.

## 2   Performance of Different Regularizations

There are only a few previous works [1, 4, 5] that study the effect of different architectures on regularization, and most of the existing works just follow the 3D CNN category. In this paper, we further explore the effect of different regularizations in Tab. 1 and find that the global receptive field has a significant impact on final performance. As shown in Fig. 4-(b) of the main paper, with the expansion of receptive field, the probability volume becomes smoother and more complete, as well as higher confidence. Compared to other regularizations, our proposed Cost Transformer (CT) obtains the best performance on DTU evaluation set [2]. However, CT inevitably suffers from increased memory consumption and inference time.

Table 1: Ablation study on the different regularizations on DTU evaluation set [2] (**lower is better**).

| Cost Regulization | Acc.($mm$) | Comp.($mm$) | Overall($mm$) | Mem.($MB$) | Time($s$) |
|---|---|---|---|---|---|
| w/o reg | 0.401 | 0.443 | 0.422 | **3283** | **0.448** |
| 2D CNN | 0.350 | 0.306 | 0.328 | 3795 | 0.488 |
| 3D CNN | 0.328 | 0.288 | 0.308 | 4017 | 0.521 |
| CT | **0.309** | **0.281** | **0.295** | 5221 | 0.786 |

## 3   Ablation Study on Hyperparameters

### 3.1   Number of Attention Blocks & Window Size in WET

As shown in Tab. 2, we explore the influence of different number of attention blocks $N$ and window size $h_{win} \times w_{win}$ in WET. We perform the ablation study on the number of attention blocks $N$ and

---

*Equal contribution.
†Corresponding author.

36th Conference on Neural Information Processing Systems (NeurIPS 2022).

model performance, memory consumption and inference time are listed in the table respectively. With the expansion of window size, the inference time reduces. Set at $N = 1$ and $h_{win} \times w_{win} = 16 \times 16$, our model achieves a balance of performance and efficiency.

Table 2: Ablation study on the number of attention blocks $N$ and the window size $h_{win} \times w_{win}$ in WET on DTU evaluation set [2] (**lower is better**).

| $N$ | $h_{win} \times w_{win}$ | Acc.($mm$) | Comp.($mm$) | Overall($mm$) | Mem.($MB$) | Time($s$) |
|---|---|---|---|---|---|---|
| 1 | $16 \times 16$ | 0.309 | **0.281** | **0.295** | 5221 | 0.786 |
| 2 | $16 \times 16$ | **0.305** | 0.284 | **0.295** | 5223 | 1.05 |
| 3 | $16 \times 16$ | 0.311 | 0.285 | 0.298 | 5225 | 1.32 |
| 1 | $8 \times 8$ | 0.312 | 0.284 | 0.298 | 5221 | 1.318 |
| 1 | $12 \times 12$ | 0.318 | 0.274 | 0.296 | 5221 | 0.922 |
| 1 | $20 \times 20$ | 0.312 | 0.287 | 0.300 | **4895** | 0.747 |
| 1 | $24 \times 24$ | 0.324 | 0.284 | 0.304 | 5039 | **0.725** |

## 3.2 Number of Attention Blocks & Window Size in CT

We further adjust the number of attention blocks $N$ and the window size $d_{win} \times h_{win} \times w_{win}$ in CT and Tab. 3 shows the evaluation results including model performance, memory consumption and inference time. With the increase of $N$, the model effect is greatly improved. Shown in Tab. 3, $N = 3$ and $d_{win} \times h_{win} \times w_{win} = 2 \times 8 \times 10$ obtain the best results without adding much memory occupancy and inference time.

Table 3: Ablation study on the number of attention blocks $N$ and the window size $d_{win} \times h_{win} \times w_{win}$ in CT on DTU evaluation set [2] (**lower is better**).

| $N$ | $d_{win} \times h_{win} \times w_{win}$ | Acc.($mm$) | Comp.($mm$) | Overall($mm$) | Mem.($MB$) | Time($s$) |
|---|---|---|---|---|---|---|
| 1 | $2 \times 8 \times 10$ | 0.315 | 0.291 | 0.303 | 5145 | **0.725** |
| 2 | $2 \times 8 \times 10$ | 0.310 | 0.289 | 0.300 | 5151 | 0.768 |
| 3 | $2 \times 8 \times 10$ | 0.309 | 0.281 | **0.295** | 5221 | 0.786 |
| 3 | $1 \times 4 \times 5$ | 0.321 | 0.284 | 0.303 | 5413 | 0.775 |
| 3 | $2 \times 4 \times 5$ | 0.316 | 0.285 | 0.301 | **5037** | 0.766 |
| 3 | $4 \times 4 \times 5$ | 0.316 | 0.284 | 0.300 | 5219 | 0.771 |
| 3 | $1 \times 8 \times 10$ | 0.314 | **0.279** | 0.297 | 5219 | 0.781 |
| 3 | $4 \times 8 \times 10$ | **0.307** | 0.284 | 0.296 | 6295 | 0.803 |

## 3.3 Loss Weights

With the weight $\lambda_1$ of cross entropy loss (CE Loss) set at 2, we conduct an ablation study on training with different weights $\lambda_2$ of geometric consistency loss (Geo Loss) in Tab. 4, Our model shows the best reconstruction performance on DTU evaluation set [2] when $\lambda_2 = 1$.

Table 4: Ablation study on the different weight $\lambda_2$ of Geo Loss. The quantitative results on DTU evaluation set [2] (**lower is better**).

| $\lambda_1$ | $\lambda_2$ | Acc.($mm$) | Comp.($mm$) | Overall($mm$) |
|---|---|---|---|---|
| 2 | 0 | 0.315 | 0.289 | 0.302 |
| 2 | 0.5 | 0.310 | 0.286 | 0.298 |
| 2 | 1 | 0.309 | **0.281** | **0.295** |
| 2 | 2 | 0.303 | 0.291 | 0.297 |
| 2 | 4 | **0.301** | 0.295 | 0.298 |

## 3.4 Number of Images & Input Resolution

We perform an ablation study on the number of input images $N$ and the input resolution $H \times W$ on DTU evaluation set [2], and the results of point clouds are summarized in Tab. 5.

Table 5: Ablation study on the number of input images $N$ and the image resolution $H \times W$ on DTU evaluation set [2] (**lower is better**).

| $N$ | $H \times W$ | Acc.($mm$) | Comp.($mm$) | Overall($mm$) | Mem.($MB$) | Time($s$) |
|---|---|---|---|---|---|---|
| 3 | $864 \times 1152$ | 0.379 | **0.245** | 0.312 | 5757 | 0.576 |
| 5 | $864 \times 1152$ | 0.309 | 0.281 | **0.295** | 5221 | 0.786 |
| 7 | $864 \times 1152$ | 0.277 | 0.361 | 0.319 | 5549 | 0.978 |
| 9 | $864 \times 1152$ | **0.273** | 0.480 | 0.377 | 5825 | 1.186 |
| 5 | $480 \times 640$ | 0.348 | 0.343 | 0.346 | **2597** | **0.266** |

## 4  More Visualization Results of Point Cloud

More visualization results of our model are shown in Fig. 1 and Fig. 2, which include all scans of DTU evaluation set [2] as well as intermediate and advanced sets of Tanks and Temples benchmark [3]. Our model demonstrates its excellent robustness and generalizability in a variety of scenarios.

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

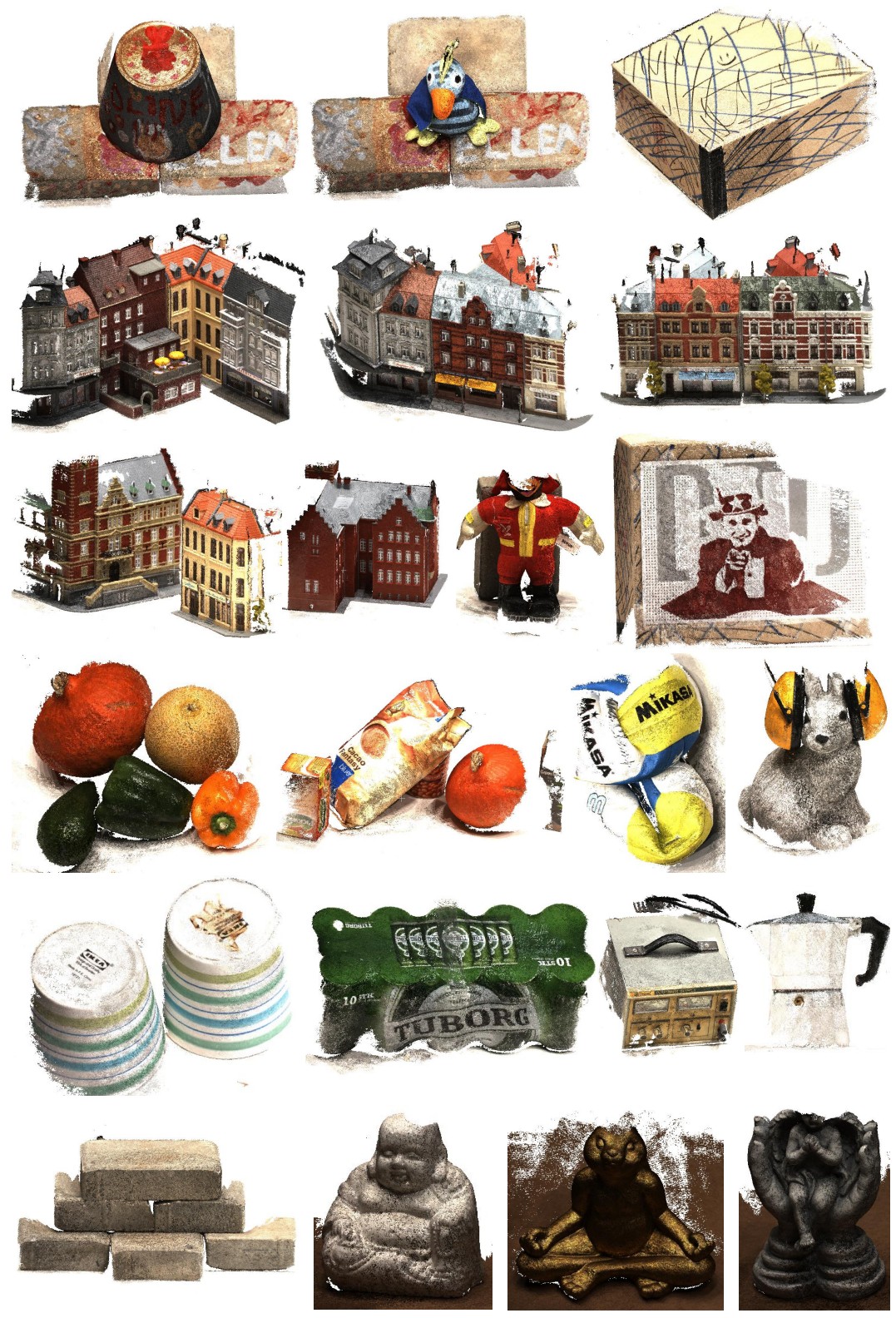

Figure 1: All point clouds reconstructed by WT-MVSNet on DTU evaluation set [2].

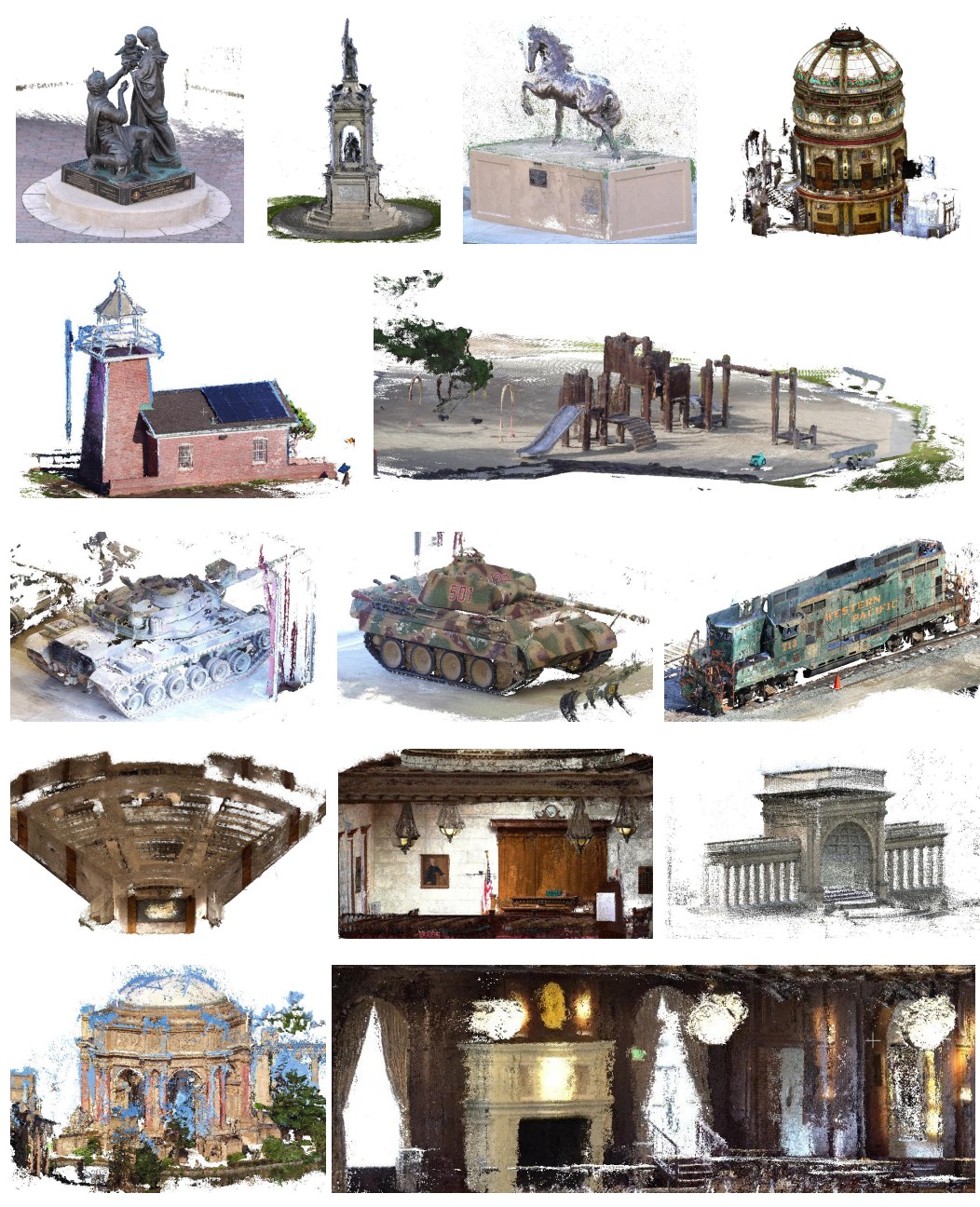

Figure 2: All point clouds reconstructed by WT-MVSNet on Tanks and Temples benchmark [3].