# OpenReview forum: "WT-MVSNet: Window-based Transformers for Multi-view Stereo"
_NeurIPS.cc/2022/Conference — NeurIPS 2022 Accept_

### Official Review · Reviewer_cEDu · 2022-07-09

**Rating:** 6
**Confidence:** 5
**Soundness:** 3 good
**Presentation:** 4 excellent
**Contribution:** 3 good

**Summary:**

This paper presents WT-MVSNet, a deep learning based multi-view stereo method. In particular, it makes uses of Swin Transformer to aggregate global context in inter- and intra-feature attentions. For inter-attention, the authors propose Window-based Epipolar Transformer (WET) to reduce matching redundancy. In the cost volume regularization phase, the authors propose to replace prior 3D U-Net based architecture with attentions in 3D, named Cost Transformer. Cross entroy and the proposed geometric loss is used as the training objectives. The experiments show that WT-MVSNet achieved state-of-the-art results and the ablation studies prove the effectivness of different proposed components.

**Questions:**

1. The authors claim that the proposed WET is robust to inaccurate pose and camera calibrations. Any experiments to prove the claim?
2. Why cross entropy loss is preferable than L1/L2 regression loss?
3. What is the runtime and memory usage for inference?

**Limitations:**

The authors have discussed about the limitations in the paper. The most notable limitation of the work is the huge memory consumption during training.

**Strengths And Weaknesses:**

## Strengths
- The paper is well written and one can follow quite smoothly.
- The way of combining (Swin) Transformer makes a lot of sense. In particular, the authors not only use Transformer to enhance 2D image feature extraction and matching, but also improve the 3D cost volume regularization with 3d attention machanism.
- The authors provide extensive evaluation results on different benchmark with state-of-the-art precision as well as detailed ablation studies.

## Weaknesses
- Overall the paper is lack of significant novelty. The work in general embeds Swin Transformer into the previously proposed Cascade MVSNet pipeline.
- For the readers who are not familiar with Swin Transformer and Cascade MVSNet, it may be a bit difficult to understand details. It would be better if the authors can provide more context on how W-MSA, SW-MSA, and Cascade MVSNet work.
- Obviously, WT-MVSNet takes so much meomry (8 x V100 with batchsize = 1) to train, which is way more than, if not all of, most of the prior works.

---

> ### Author Response · Authors · 2022-08-02
> **Authors response**
>
> We thank the reviewer for the insightful and thorough review. We will further incorporate other suggestions of the reviewer in the next version. In the following, we address the main concerns raised in this review.
>
> ***Q: Overall the paper is lack of significant novelty.***
> A: We think there are three novel modules in our paper. First, WET is the first attempt among learning-based MVS methods to utilize the window-based transformers for handling feature matching problem, taking into account the inaccurate camera calibration, which is inspiring and novel. Second, CT is the first to use transformers for aggregating global information in cost regularization, making depth maps smoother and more complete. Third, Geo Loss is the first to introduce reprojection errors into the loss function in learning-based MVS, which is well-suited and effective for 3D reconstruction.
>
> ***Q: It would be better if the authors can provide more context on how W-MSA, SW-MSA, and Cascade MVSNet work.***
> A: Thanks for your suggestions, and we will add more context about W-MSA, SW-MSA and CasMVSNet work in the next version.
>
> ***Q: WT-MVSNet takes so much meomry to train.***
> A: We guess the reviewer may think our method costs 32*G* memory in training phase, but actually it costs 13*G* as described in Ln-217.
>
> ***Q: Experiments to prove the claim that the WET is robust to inaccurate pose and camera calibration.***
> A: This is a great question. We tried to perform experiments in the DTU test set, but we found it hard to properly simulate the inaccurate calibration because the random noise doesn't conform with the real calibration error and makes the final results strange and bad (for both our method and CasMVSNet). In that context, we think the results on the Tanks and Temples benchmark are good proof to prove the robustness of our method to inaccurate poses and calibration. More specifically, the poses and calibration of Tanks and Temples are obtained by COLMAP, which are imperfect but conform with the real error. Taking such poses and calibration as inputs, our method achieves much better results on both intermediate and advanced sets than existing methods. We also believe that more experimental results would be helpful to improve our paper, and we will continue to design better experiments to support our contribution.
>
> ***Q: Why cross entropy loss is preferable than L1/L2 regression loss?***
> A: As we treat the MVS as a feature matching task, we seek to find the best matches in source views for each pixel in the reference view. In such problem, we think the classification loss function is more suitable and effective.
> More specifically, the L1 and L2 losses only concentrate on the expectations of the probability volume and may have a significant variance, while the cross entropy loss constrains both the expectations and distribution of the probability volume.
>
> ***Q: What is the runtime and memory usage for inference?***
> A: We give the runtime and memory for inference in supplementary material, which are 0.786*s* and 5221*MB* respectively. We will add more details in the final version.

---

> > ### Comment · Reviewer_cEDu · 2022-08-08
> > **Thanks for authors' detailed response.**
> >
> > Thanks for authors' detailed response. I will keep my original rating.

---

### Official Review · Reviewer_rrwL · 2022-07-09

**Rating:** 7
**Confidence:** 4
**Soundness:** 3 good
**Presentation:** 3 good
**Contribution:** 3 good

**Summary:**

This paper proposes a transformer-based architecture for multi-view stereo. Main contributions include inter-frame and inter-frame attention mechanisms designed to leverage epipolar constraints. The method is simple and performs very well on standard benchmarks. Ablations show that the proposed components are the contributing factors for strong performance.

**Questions:**

How is depth filtering and post-processing performed when submitting to evaluation benchmarks?
How is the cost volume computed?
How is the geometric consistency loss able to properly handle outliers caused by occlusions?


**Limitations:**

Yes limitations are adequately addressed.

**Strengths And Weaknesses:**

Strengths:
* The network architecture is simple - especially compared to other top performing neural network architectures for multi-view stereo.
* The epipolar-window transformer is a novel and interesting contribution
* Results on the DTU and Tanks & Temples dataset are very good.
* The visualization of the probability volume is interesting and not something that has been typically shown with prior MVS nets.


Weaknesses:
* Some components of the network are not properly explained. For example, nowhere in the paper is it stated how the cost volume V is computed (I assume it is constructed using the same method as CasMVSNet but this is not explicitly stated). There is also no mention of filtering or post-processing in the paper. This is an important step of MVS systems which can significantly impact performance.
* This is not a weakness of this work in particular, but many deep MVS papers in general. There are no results presented on the ETH MVS dataset (either low-res or high-res). Deep MVS networks perform very well on DTU and Tanks & Temples but much less so on ETH. I think it would be helpful to show results on datasets where deep learning has been somewhat less successful compared to classical MVS approaches in order to better assess progress, rather than just showing results on datasets where deep learning is successful. I would not consider non-SOTA results on this benchmark to be a weakness of the paper.
* Memory and timing are only mentioned in the supplemental. Basic time / memory requirements should be given in the main body of the paper.

Not major weaknesses, but some related work that I think should be discussed in more detail in the relevant sections
*  The CT Architecture is very closely related to the Dense Prediction Transformer [20] (main difference being 2D vs 3D processing). I think DPT should be referenced in Sec. 3.3.
* Many ideas presented in this work are closely related to GMFlow (Xu et al, CVPR 2022) - such as inter / intra frame attention for obtaining a better cost volume.

---

> ### Author Response · Authors · 2022-08-02
> **Authors response**
>
> We thank the reviewer for the insightful and thorough review. We will further incorporate other suggestions of the reviewer in the next version. In the following, we address the main concerns raised in this review.
>
> ***Q: Some components of the network are not properly explained, including how to compute the cost volume $V$ and how to perform post-processing.***
> A: Thanks for your suggestions, and we will add more details in the next version accordingly.
>
> ***Q: There are no results presented on the ETH3D dataset.***
> A: For ETH3D, we fail to find enough results from previous SOTA methods for comparison. But we would like to thank the reviewer for this suggestion and will include the results in a later version.
>
> ***Q: Memory and timing are only mentioned in the Supplementary Material. Basic time / memory requirements should be given in the main body of the paper.***
> A: As described in the last sentence of Sec. 4.1, our model typically takes 15 hours and occupies 13*GB* of each GPU's RAM with the batch size being set to 1 on 8 Tesla V100 GPUs during training phase. The memory and time during inference phase are shown in supplementary material. We will add the memory and runtime of inference phase to Sec. 4.1 in our next version.
>
> ***Q: The CT Architecture is very closely related to the Dense Prediction Transformer [20] (main difference being 2D vs 3D processing). I think DPT should be referenced in Sec. 3.3.***
> A: Thanks for your advice, and we will cite DPT in the next version.
>
> ***Q: Many ideas presented in this work are closely related to GMFlow (Xu et al, CVPR 2022) - such as inter / intra frame attention for obtaining a better cost volume.***
> A: We think the GMFlow is a concurrent work. Although it was accepted by CVPR'22, their paper didn't be made public before we submitted our paper.
>
> ***Q: How is the post-processing performed when submitting to evaluation benchmarks?***
> A: As the post-processing is a critical part in MVS pipeline, we follow the previous methods [3, 7, 19] to fuse the point clouds for a fair comparison. More specifically, we follow the fusion method in CasMVSNet [7] when testing on DTU dataset, and follow the dynamic fusion method used in TransMVSNet [3] and UniMVSNet [19] when submitting to Tanks and Temples benchmark. We'll add more details in our final version.
>
> ***Q: How is the cost volume computed?***
> A: As mentioned in Ln-249, we construct cost volume by computing feature correlation by following PatchmatchNet [26] and TransMVSNet [3] as this formula is more consistent with feature matching. We will add more details of this part in our final version.
>
> ***Q: How is the geometric consistency loss able to properly handle outliers caused by occlusions?***
> A: As occlusion is an inherent problem in MVS task, the corresponding outliers can be detected by checking the cross-view consistency as done in Geo Loss. More specifically, the occluded outliers usually possess large reprojection errors, which are harmful samples in the training phase. To depress the influence of such noisy pixels, we filter the occluded pixels which don't meet the reprojection threshold when computing Geo Loss.

---

### Official Review · Reviewer_h2KQ · 2022-07-10

**Rating:** 7
**Confidence:** 4
**Soundness:** 3 good
**Presentation:** 3 good
**Contribution:** 3 good

**Summary:**

The paper proposes a new learned MVS method based on MVSNet pipeline.
The main contribution includes the introduction of window-based transformers to exchange the information among different views and a Geo Loss to consider the supervision signal of multiple views.
The experiements show the proposed method achieves a new state-of-the-art and ranks 1st on the tanks and temple dataset.

**Questions:**

The differentiable warping part in MVSNet actually implicitly tries to find the best matches among epipolar lines, so the window-based epipolar inter-attention module seems do the same things. How do the authors think about this?

**Limitations:**

The authors discuss the limitation briefly and the work does not have potential negative societal impact.

**Strengths And Weaknesses:**

Strengths
1. The paper is generally well written. All the figures and equations are clear.
2. It is novel to introduce transformer in the learned MVS method to exchange the information among different views before differentiable warping.
3. The Geo Loss is novel and reasonable to condiser the supervision signal of multiple views.
4. The overal scores rank 1st on the tanks and temple dataset.

Weaknesses
1. Although the framework of MVSNet and NeRF based methods are different, they have the same inputs and can output put the 3D representation of the objects. However, the authors do not mention NeRF based methods in the whole paper.
2. The memory cost of the transformer is large, so the proposed method cannot deal with high resolution inputs.

---

> ### Author Response · Authors · 2022-08-02
> **Authors response**
>
> We thank the reviewer for the insightful and thorough review. We will further incorporate other suggestions of the reviewer in the next version. In the following, we address the main concerns raised in this review.
>
> ***Q: Why not mention and compare NeRF based methods in the whole paper?***
> A: This is a good question and we will add more discussion with NeRF-based methods in our next version. For now, while the inputs of MVS nets and NeRF are the same, there is a significant difference between these methods. For example, NeRF mainly focuses on novel view synthesis, and its 3D reconstruction results are relatively poor (see results in Tab. 1 of NeuS). More specifically, the reconstruction results of NeRF on DTU (chamfer distance equals mean error) are far inferior to MVS nets.
>
> ***Q: The differentiable warping part in MVSNet actually implicitly tries to find the best matches among epipolar lines, so the window-based epipolar inter-attention module seems do the same things. How do the authors think about this?***
> A: This is an interesting question. First, the differentiable warping itself doesn't perform the feature matching, it only warps the features from source views to the reference view for constructing the cost volume. MVSNet tries to find the best matches by computing the variance among different features (some methods compute the correlation instead of the variance). Second, our window-based epipolar inter-attention module aims to improve the feature quality to enhance the feature matching. It is worth noting that the differentiable warping part and the inter-attention module are not in conflict, they work together in our whole pipeline.

---

### Official Review · Reviewer_nW8B · 2022-07-12

**Rating:** 6
**Confidence:** 4
**Soundness:** 2 fair
**Presentation:** 3 good
**Contribution:** 3 good

**Summary:**

The paper considers the problem of Multi-View Stereo reconstruction from images. It proposes a transformer-based approach, where specialized transformers are introduced for epipolar-guided stereo matching (in a region around the epipolar line to account for inaccuracies in camera calibration and poses), for aggregating costs, and for regularization in the cost volume. The proposed approach represents the current state-of-the-art on both the intermediate and hard scenes of the Tanks & Temples dataset.

**Questions:**

I will consider raising my rating if the weaknesses W1-W5 are successfully addressed in a rebuttal. Addressing W2 is particularly important.

**Limitations:**

Limitations are reasonably addressed.

**Strengths And Weaknesses:**

Strength:
S1) The proposed approach achieves state-of-the-art results on the challenging Tanks & Temples dataset (both for the intermediate and the hard categories). To me, the results on Tanks & Temples are one of the main strengths of the paper.

S2) The proposed pipeline is technically sound and well-engineered. It is clearly described.

Weaknesses:
W1) The paper states that "In order to introduce epipolar constraints into attention-based feature matching while maintaining robustness to camera pose and calibration inaccuracies, we develop a Window-based Epipolar Transformer (WET), which matches reference pixels and source windows near the epipolar lines." It claims that it introduces "a window-based epipolar Transformer (WET) for enhancing patch-to-patch matching between the reference feature and corresponding windows near epipolar lines in source features". To me, taking a window around the epipolar line into account seems like an approximation to estimating the uncertainty region around the epipolar lines caused by inaccuracies in calibration and camera pose and then searching within this region (see [Förstner & Wrobel, Photogrammetric Computer Vision, Springer 2016] for a detailed derivation of how to estimate uncertainties). Is it really valid to claim this part of the proposed approach as novel?

W2) I am not sure how significant the results on the DTU dataset are:
a) The difference with respect to the best performing methods is less than 0.1 mm (see Tab. 1). Is the ground truth sufficiently accurate enough that such a small difference is actually noticeable / measurable or is the difference due to noise or randomness in the training process?
b) Similarly, there is little difference between the results reported for the ablation study in Tab. 4. Does the claim "It can be seen from the table that our proposed modules improve in both accuracy and completeness" really hold?
Why not use another dataset for the ablation study, e.g., the training set of Tanks & Temples or ETH3D?

W3) I am not sure what is novel about the "novel geometric consistency loss (Geo Loss)". Looking at Eq. 10, it seems to simply combine a standard reprojection error in an image with a loss on the depth difference.  I don't see how Eq. 10 provides a combination of both losses.

W4) While the paper discusses prior work in Sec. 2, there is mostly no mentioning on how the paper under review is related to these existing works. In my opinion, a related work section should explain the relation of prior work to the proposed approach. This is missing.

W5) There are multiple parts in the paper that are unclear to me:
a) What is C in line 106? The term does not seem to be introduced.
b) How are the hyperparameters in Sec. 4.1 chosen? Is their choice critical?
c) Why not include UniMVSNet in Fig. 5, given that UniMVSNet also claims to generate denser point clouds (as does the paper under review)?
d) Why use only N=5 images for DTU and not all available ones?
e) Why is Eq. 9 a reprojection error? Eq. 9 measures the depth difference as a scalar and no projection into the image is involved. I don't see how any projection is involved in this loss.

Overall, I think this is a solid paper that presents a well-engineered pipeline that represents the current state-of-the-art on a challenging benchmark. While I raised multiple concerns, most of them should be easy to address. E.g., I don't think that removing the novelty claim from W1 would make the paper weaker. The main exception is the ablation study, where I believe that the DTU dataset is too easy to provide meaningful comparisons (the relatively small differences might be explained by randomness in the training process.

The following minor comments did not affect my recommendation:
* References are missing for Pytorch and the Adam optimizer.

**Post-rebuttal comments**

Thank you for the detailed answers. Here are my comments to the last reply:

>  Q: Relationship to prior work.

Thank you very much, this addresses my concern.

>  A: Fig. 5 is not used to claim our method achieves the best performance among all the methods in terms of completeness, it actually indicates that our proposed method could help reconstruct complete results while keeping high accuracy (Tab. 1) compared with our baseline network [7] and the most relevant method [3]. In that context, we not only consider the quality of completeness but also the relevance to our method to perform comparison in Fig. 5.

As I understand lines 228-236 in the paper, in particular "The quantitative results of DTU evaluation set are summarized in Tab. 1, where Accuracy and Completeness are a pair of official evaluation metrics. Accuracy is the percentage of generated point clouds matched in the ground truth point clouds, while Completeness measures the opposite. Overall is the mean of Accuracy and Completeness. Compared with the other methods, our proposed method shows its capability for generating denser and more complete point clouds on textureless regions, which is visualized in Fig. 5.", the paper seems to claim that the proposed method generates denser point clouds. Maybe this could be clarified?

>  A: As a) nearly all the learning-based MVS methods (including ours) take the DTU as an important dataset for evaluation, b) the GT of DTU is approximately the most accurate GT we can obtain (compared with other datasets), c) the final results are the average across 22 test scans, we think that fewer errors could indicate better performance. However, your point about the accuracy of DTU GT is enlightening, and we think it's valuable future work.

This still does not address my concern. My question is whether the ground truth is accurate enough that we can be sure that the small differences between the different components really comes from improvements provided by adding components. In this context, stating that "the GT of DTU is approximately the most accurate GT we can obtain (compared with other datasets)" does not answer this question as, even though DTU has the most accurate GT, it might not be accurate enough to measure differences at this level of accuracy (0.05 mm difference). If the GT is not accurate enough to differentiate in the 0.05 mm range, then averaging over different test scans will not really help. That "nearly all the learning-based MVS methods (including ours) take the DTU as an important dataset for evaluation" does also not address this question. Since the paper claims improvements when using the different components and uses the results to validate the components, I do not think that answering the question whether the ground truth is accurate enough to make these claims in future work is really an option. I think it would be better to run the ablation study on a dataset where improvements can be measured more clearly.

**Final rating**

I am inclined to keep my original rating ("6: Weak Accept: Technically solid, moderate-to-high impact paper, with no major concerns with respect to evaluation, resources, reproducibility, ethical considerations."). I still like the good results on the Tanks & Temples dataset and believe that the proposed approach is technically sound. However, I do not find the authors' rebuttals particularly convincing and thus do not want to increase my rating. In particular, I still have concerns about the ablation study as I am not sure whether the ground truth of the DTU dataset is accurate enough that it makes sense to claim improvements if the difference is 0.05 mm or smaller. Since this only impacts the ablation study, it is also not a reason to decrease my rating.

---

> ### Author Response · Authors · 2022-08-02
> **Authors response**
>
> We thank the reviewer for the insightful and thorough review. We will further incorporate other suggestions of the reviewer in the next version. In the following, we address the main concerns raised in this review.
>
> ***Q: Is it really valid to claim WET of the proposed approach as novel?***
> A: To the best of our knowledge, our WET is the first attempt among learning-based MVS methods to utilize the window-based transformer for handling the feature matching problem, taking into account the inaccurate camera calibration. Even though a similar ideology has been explored in classical methods, we think WET is an inspiring and novel design for learning-based methods.
>
> ***Q: Is the ground truth of DTU dataset accurate enough to measure the small error?***
> A: This is a good question. There are two reasons why we think the results on DTU are convincing. First, the ground truth of DTU is pretty accurate (rendered using high-quality 3D mesh [8]), even though the average scores among 22 testing scans of WT-MVSNet just surpass the existing methods slightly in terms of the quantitative results, the qualitative results vary widely (as shown in Fig. 5), so we think the superior performance of WT-MVSNet is persuasive. Second, as randomness and noise indeed exist in our experiments, we trained our method for 3 times with the same random seeds to suppress the influence of randomness in training phase.
>
> ***Q: Why not use another dataset for the ablation study, e.g. Tanks and Temples or ETH3D?***
> A: Based on the analysis above, we think the reported results in Tab. 4 vary widely (see Tab. 1 and Tab. 4), and each proposed module brings significant improvement in the final results. As nearly all the learning-based MVS methods perform the ablation study on DTU test set, we follow this paradigm. Nevertheless, we believe that using more datasets in the ablation study is a good idea, we'll try and record more results in our final version.
>
> ***Q: What is novel about Geo Loss? How the Eq. 10 provides a combination of both reprojection errors losses?***
> A: For the first question, our method is the first to introduce reprojection errors into the training phase in learning-based MVS tasks, which is well-suited and effective for the 3D reconstruction problem [reviewer h2KQ]. For the second question, we guess the reviewer may be confused by why Eq. 10 represents a combination of two reprojection errors. Eq. 10 is a sum of Eq. 8 and Eq. 9 over the valid pixels, which indicates two reprojection errors. More specifically, Eq. 8 is a classical reprojection error and Eq. 9 is a relative depth error, which also reflects the reprojection error and is widely called reprojection error in MVS.
>
> ***Q: The related work section should explain the relation of prior work to the proposed approach. This is missing.***
> A: Thanks for your suggestions, and we will add more details in our next version accordingly.
>
> ***Q: What is C in Ln-106?***
> A: The C is the number of feature channels, and we will make the description clearer in our final version.
>
> ***Q: How are the hyperparameters in Sec. 4.1 chosen? Is their choice critical?***
> A: For most hyperparameters in Sec. 4.1, we follow the setting in previous work [3,7], our new hyperparameters (number of attention blocks, window size, etc) are explained in supplementary material. According to the results in Sec. 3 of supp, the choice of some hyperparameters is critical (e.g., resolution, view number, etc).
>
> ***Q: Why not include UniMVSNet in Fig. 5, given that UniMVSNet also claims to generate denser point clouds?***
> A: There are two reasons why we chose TransMVSNet [3] instead of UniMVSNet [19] to compare in Fig. 5. First, the performance of TransMVSNet is better than UniMVSNet on DTU test set. Second, TransMVSNet is more relevant to our method (both leverage transformers). Additionally, we chose to compare with CasMVSNet [7] because our method is developed upon it.
>
> ***Q: Why use only N=5 images for DTU and not all available ones?***
> A: This is an interesting question. To illustrate the influence of $N$, we perform an ablation study in Tab. 5 of the supplementary material. As shown in the results, the optimal $N$ for DTU dataset is 5 in our case. Here we give our analysis for this ablation study.
> Since we are performing experiments on DTU test set, where the camera distribution is quite sparse, the problem of occlusion becomes severer as the number of input views increases. Similar results are also observed in TransMVSNet [3] and PatchmatchNet [26].
>
> ***Q: Why is Eq. 9 a reprojection error? Eq. 9 measures the depth difference as a scalar and no projection into the image is involved.***
> A: As described in the above question, Eq. 9 calculates the relative depth error between the back-projected pixel $\mathbf{p}^{''}$ and the original pixel $\mathbf{p}$ on the estimated depth map $\mathbf{D}_{0}$ , which also reflects the reprojection error and is widely called reprojection error in MVS.

---

> > ### Comment · Reviewer_nW8B · 2022-08-07
> > **Re: Authors response - 1**
> >
> > Thank you very much for the detailed answers and comments. Please see my comments and follow-up questions below.
> >
> > >  To the best of our knowledge, our WET is the first attempt among learning-based MVS methods to utilize the window-based transformer for handling the feature matching problem, taking into account the inaccurate camera calibration. Even though a similar ideology has been explored in classical methods, we think WET is an inspiring and novel design for learning-based methods.
> >
> > Please let me clarify. My understanding of the sentence (and similar sentences) " In order to introduce epipolar constraints into attention-based feature matching while maintaining robustness to camera pose and calibration inaccuracies, we develop a Window-based Epipolar Transformer (WET), which matches reference pixels and source windows near the epipolar lines." is that a main part of the novelty lies in matching near the epipolar line (as opposed to just along the epipolar line). My argument is that searching close to the epipolar line is nothing new and that the proposed approach only represents an approximation to properly defining the search space based on the uncertainty in the camera poses and intrinsics (which are not really taken into account as the search region does not seem to depend on them). Thus my question whether this really can be claimed as a novelty. I guess this could be resolved by stating that WET is motivated by the known fact that due to pose and calibration inaccuracies, better results can be obtained by searching in a region around the epipolar line.
> >
> > > This is a good question. There are two reasons why we think the results on DTU are convincing. First, the ground truth of DTU is pretty accurate (rendered using high-quality 3D mesh [8]), even though the average scores among 22 testing scans of WT-MVSNet just surpass the existing methods slightly in terms of the quantitative results, the qualitative results vary widely (as shown in Fig. 5), so we think the superior performance of WT-MVSNet is persuasive. Second, as randomness and noise indeed exist in our experiments, we trained our method for 3 times with the same random seeds to suppress the influence of randomness in training phase.
> >
> > I find this answer unconvincing, as it depends on what "pretty accurate" means, which was my question in the first place. Is the combination of the errors in the ground truth mesh (which according to the IJCV paper seems to be in the order of 0.1 mm), camera poses (according to the paper around 0.05 mm), and camera intrinsics small that differences of 0.05 mm or less really indicate better performance?
> >
> > From the qualitative results in Fig. 5, I can only say which method produces more complete results, not whether one method is more accurate than another.
> >
> > I am not sure why fixing the random seed and training the method 3 times with the same seed helps to suppress the influence of randomness. Wouldn't it make more sense to use 3 different random seeds?
> >
> > > Based on the analysis above, we think the reported results in Tab. 4 vary widely (see Tab. 1 and Tab. 4), and each proposed module brings significant improvement in the final results. As nearly all the learning-based MVS methods perform the ablation study on DTU test set, we follow this paradigm. Nevertheless, we believe that using more datasets in the ablation study is a good idea, we'll try and record more results in our final version.
> >
> > My question is whether the improvements reported in Tab. 4 are large enough with respect to the accuracy of the ground truth, etc. that significant improvements can be claimed. I don't see how this comment answers my question.
> >
> > > Thanks for your suggestions, and we will add more details in our next version accordingly.
> >
> > Would it be possible to summarize the details here? To me, explaining the relationship to prior work is an important part of a paper (hence the discussion above about the novelty of the concept of matching to larger regions). I had hoped for a brief summary when I asked for this.

---

> > > ### Author Response · Authors · 2022-08-09
> > > **Re: Reviewer comments**
> > >
> > > Thank you very much for the insightful and detailed comments, many of them could help us a lot.
> > >
> > > ***Q: Relationship to prior work.*** \
> > > A: a) As described in Sec. 2.1, one of the main distinctions between different learning-based MVS methods lies in the cost regularization, which can be classified as RNN-based [25, 27, 28, 30, 34] and 3D CNN-based methods [2, 7, 31, 36, 37].
> > > However, neither of these kinds of methods utilize a global receptive field to perform regularization.
> > > Unlike these methods, our proposed CT first uses 3D window-based transformers for aggregating global information in cost regularization, making depth maps smoother and more complete.
> > >
> > > b) As described in Sec. 2.2, some recent works [10,21,23] have demonstrated the significance of introducing transformers to enhance feature matching. Treating MVS as a feature matching task, recent learning-based MVS networks [3, 32] also attempt to use transformers to enhance the one-to-many matching.
> > > TransMVSNet [3] first introduces transformers into MVS task, which matches each reference pixel with the whole source images without using the epipolar constraints. MVS2D [32] uses the attention mechanism to perform point-to-line matching while ignoring the inaccurate camera calibrations.
> > > To solve such problems, we propose the WET that utilizes the window-based transformers for enhancing patch-to-patch matching, taking into account the epipolar constraint and the inaccurate camera calibration.
> > > Compared with TransMVSNet, our WET uses a much smaller matching space and costs less computation; compared with MVS2D, WET is more robust to the imperfect camera calibrations and poses.
> > >
> > > ***Q:  The conclusion from Fig. 5 is that "Compared with our baseline [7] and TransMVSNet [3], our WT-MVSNet obtains a more complete point cloud shown in the red bounding boxes." Looking at Tab. 1 in the paper under review and Fig. 6 in the UniMVSNet paper, UniMVSNet produces more complete results that TransMVSNet and would thus seem a better baseline. Similarly, GBi-Net would make a better baseline.*** \
> > > A: Fig. 5 is not used to claim our method achieves the best performance among all the methods in terms of completeness, it actually indicates that our proposed method could help reconstruct complete results while keeping high accuracy (Tab. 1) compared with our baseline network [7] and the most relevant method [3]. In that context, we not only consider the quality of completeness but also the relevance to our method to perform comparison in Fig. 5.
> > >
> > > ***Q: My question is whether the improvements reported in Tab. 4 are large enough with respect to the accuracy of the ground truth, etc. that significant improvements can be claimed. & Is the combination of the errors in the ground truth mesh (which according to the IJCV paper seems to be in the order of 0.1 mm), camera poses (according to the paper around 0.05 mm), and camera intrinsics small that differences of 0.05 mm or less really indicate better performance?*** \
> > > A: As a) nearly all the learning-based MVS methods (including ours) take the DTU as an important dataset for evaluation, b) the GT of DTU is approximately the most accurate GT we can obtain (compared with other datasets), c) the final results are the average across 22 test scans, we think that fewer errors could indicate better performance. However, your point about the accuracy of DTU GT is enlightening, and we think it's valuable future work.
> > >
> > > ***Q: Wouldn't it make more sense to use 3 different random seeds?*** \
> > > A: The main reason that we use the same seeds is to ensure that our results could be reproduced (as done in previous works [3,7,19,27]), so our last response about suppressing the influence of randomness may be inaccurate.

---

> > ### Comment · Reviewer_nW8B · 2022-08-07
> > **Re: Authors response - 2**
> >
> > Thank you very much for the detailed answers and comments. Please see my comments and follow-up questions below.
> >
> > > There are two reasons why we chose TransMVSNet [3] instead of UniMVSNet [19] to compare in Fig. 5. First, the performance of TransMVSNet is better than UniMVSNet on DTU test set. Second, TransMVSNet is more relevant to our method (both leverage transformers). Additionally, we chose to compare with CasMVSNet [7] because our method is developed upon it.
> >
> > The conclusion from Fig. 5 is that "Compared with our baseline [ 7] and TransMVSNet [ 3], our WT-MVSNet obtains a more complete point cloud shown in the red bounding boxes." Looking at Tab. 1 in the paper under review and Fig. 6 in the UniMVSNet paper, UniMVSNet produces more complete results that TransMVSNet and would thus seem a better baseline. Similarly, GBi-Net would make a better baseline.

---

### Meta-Review · Area_Chair_c359 · 2022-08-23

**Recommendation:** Accept
**Confidence:** Certain

**Metareview:**

The paper is concerned with multi-View Stereo reconstruction from images with several transformers for specific subtasks. SOTA performance is attained. Reviewers acknowledge a technically sound pipeline. The writing is also clear and limitations are addressed. All reviewers recommend the paper for acceptance and so do I.
Nevertheless, the authors must include the feedback provided by reviewers e.g. on possibly limited significance of 0.05 mm better results on DTU is significant or not given accuracy limitations of the ground truth. Also connections to pointed out related work must be discussed.

**Award:**

No

---

### Decision · Program_Chairs · 2022-09-14

Accept